# FIREACT: TOWARD LANGUAGE AGENT FINE-TUNING

## ABSTRACT

Recent efforts have augmented language models (LMs) with external tools or environments, leading to the development of *language agents* that can reason and act. However, most of these agents rely on few-shot prompting techniques with off-the-shelf LMs. In this paper, we investigate and argue for the overlooked direction of fine-tuning LMs to obtain language agents. Using a setup of question answering (QA) with a Google search API, we explore a variety of base LMs, prompting methods, fine-tuning data, and QA tasks, and find language agents are consistently improved after fine-tuning their backbone LMs. For example, fine-tuning Llama2-7B with 500 agent trajectories generated by GPT-4 leads to a 77% HotpotQA performance increase. Furthermore, we propose FireAct, a novel approach to fine-tuning LMs with trajectories from multiple tasks and prompting methods, and show having more diverse fine-tuning data can further improve agents. Along with other findings regarding scaling effects, robustness, generalization, efficiency and cost, our work establishes comprehensive benefits of fine-tuning LMs for agents, and provides an initial set of experimental designs, insights, as well as open questions toward language agent fine-tuning.

## 1 INTRODUCTION

Recent work has explored grounding language models (LMs; Brown et al., 2020; Chowdhery et al., 2022; Touvron et al., 2023a) to interact with external tools or environments, leading to a new class of *language agents* (Nakano et al., 2021; Yao et al., 2022b; Park et al., 2023) that could obtain new knowledge from environmental feedback, make sequential decisions via language reasoning, and improve task solving using self-reflection (Shinn et al., 2023; Wang et al., 2023a). Beyond research, industrial developments such as ChatGPT Plugins (OpenAI, 2023c) have indicated the great potential of language agents for real-world applications.

So far, most language agents prompt off-the-shelf LMs for convenience and flexibility. However, existing LMs were not developed for agentic usecases (e.g., generating actions or self-evaluations), for which few-shot prompting only offers limited learning support. As a result, most LMs have poor performance and robustness when used for agents, and some advanced agents (Yao et al., 2023; Wang et al., 2023a) can only be supported by GPT-4 (OpenAI, 2023b), resulting in high costs and latencies, along with issues like controllability and reproducibility.

Fine-tuning is an appropriate solution for these issues: it has been shown that fine-tuned smaller LMs could outperform prompted larger LMs for specific reasoning (Zelikman et al., 2022; Huang et al., 2022a) and acting (Yao et al., 2022b) needs, while enjoying reduced inference time and expense. But the study of LM fine-tuning for agents has been very limited, despite the large amount of studies around language agents and LM fine-tuning respectively (Figure 1). Only a few prior works have fine-tuned LMs for web navigation (Nakano et al., 2021; Yao et al., 2022a) or API tool use (Schick et al., 2023; Patil et al., 2023; Qin et al., 2023), with preliminary scaling analysis specific to a type of models (Yao et al., 2022b; Schick et al., 2023; Nakano et al., 2021).

In this work, we take an initial step toward a more systematic study of language agent fine-tuning. We propose FireAct, a novel way to fine-tune LMs with agent trajectories generated from multiple tasks and prompting methods, and unified in the ReAct (Yao et al., 2022b) format (Figure 2). We implement FireAct using open-domain question answering (QA) tasks with access to a Google search API, and GPT-4 (OpenAI, 2023b) for fine-tuning data generation. By thoroughly investigating a variety of base LMs (OpenAI, 2023a; Touvron et al., 2023a; Rozière et al., 2023), prompting

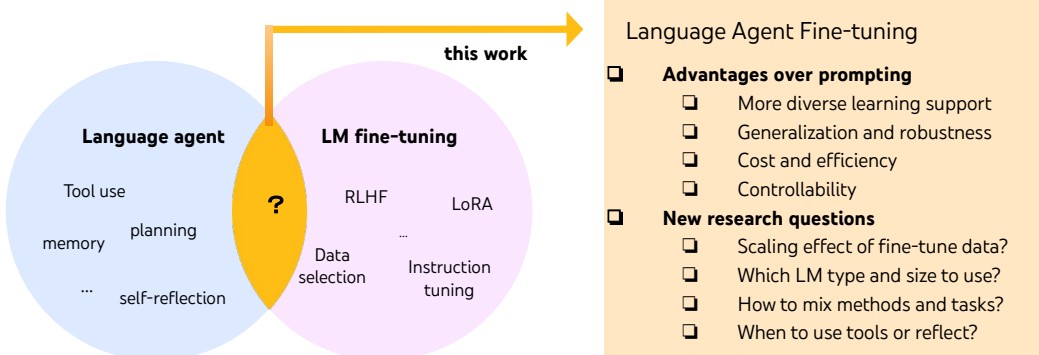

Figure 1: While language agents and language model fine-tuning are both popular topics, their intersection is understudied. This work takes an initial step to show multiple advantages of fine-tuning LMs for agentic uses, and opens up various new questions toward language agent fine-tuning.

methods (Yao et al., 2022b; Wei et al., 2022b; Shinn et al., 2023), fine-tuning data, and tasks (Yang et al., 2018; Press et al., 2022; Hendrycks et al., 2021; Geva et al., 2021), our experiments illustrate various advantages of fine-tuning and the importance of fine-tuning data diversity. For example, while few-shot `ReAct` prompting GPT-3.5 on HotpotQA achieves an exact match (EM) score of 31.4, fine-tuning with 500 `ReAct` trajectories improves the EM to 39.2 (25% increase), and fine-tuning with a mix of `ReAct` and CoT trajectories further improves the EM to 41.0 (31% increase). Furthermore, fine-tuning reduces inference time by 4x, and improves performances by 64% in face of distracting tool outputs. Such benefits can be even more visible for smaller open-source LMs where few-shot prompting performs poorly, e.g., fine-tuning Llama2-7B (Touvron et al., 2023a) leads to a 77% EM increase on HotpotQA.

Besides showcasing these benefits, our experiments also explore complex interactions among various factors of fine-tuning and provide actionable insights for practitioners. As for the base LM, we find GPT-3.5 significantly outperforms other open-source LMs when fine-tuning with less than 500 samples, but the gap can be gradually caught up by scaling to more fine-tuning samples. As for the prompting methods to generate fine-tuning data, we find different LMs benefit from different mix ratios, and present trajectory statistics and oracle analyses for further understanding. As for the tasks to generate fine-tuning data, our preliminary results show that adding a task might not improve downstream performances on significantly different tasks, but also does not hurt performances. This suggests the potential for massive multi-task fine-tuning to obtain a single LM as the agent backbone for various applications. Along with various other findings, discussions, and the release of `FireAct` code, data, and model checkpoints, we hope our work ignites and inspires future efforts toward more capable and useful fine-tuned language agents.

## 2   RELATED WORK

**Language agents**. Language agents (Weng, 2023; Wang et al., 2023b) represent an emerging kind of AI systems that use language models (LMs) to interact with the world. While earliest language agents simply used LMs to generate action commands (Nakano et al., 2021; Huang et al., 2022b; Ahn et al., 2022; Schick et al., 2023), learning direct observation-action mappings from few-shot demonstrations is challenging when the domain is complex or involves long-horizon activities. ReAct (Yao et al., 2022b) proposed to use LMs to generating both reasoning traces (Wei et al., 2022b; Nye et al., 2021; Kojima et al., 2022) and actions, so that reasoning can flexibly guide, track, and adjust acting, leading to substantial improvements over act-only methods. Follow up work has applied LM-based reasoning for more purposes in agent design, such as reflection (Shinn et al., 2023; Park et al., 2023), planning (Yao et al., 2023; Dagan et al., 2023; Liu et al., 2023a), program synthesis (Liang et al., 2023; Wang et al., 2023a), etc. The forms of external grounding have also diversified, ranging from digital games (Huang et al., 2022b; Wang et al., 2023a), APIs ("tools"; Schick et al., 2023; Patil et al., 2023; Qin et al., 2023), webpages (Yao et al., 2022a; Deng et al., 2023; Zhou et al., 2023b), to physical (Bharadhwaj et al., 2023; Vemprala et al., 2023; Driess et al.,

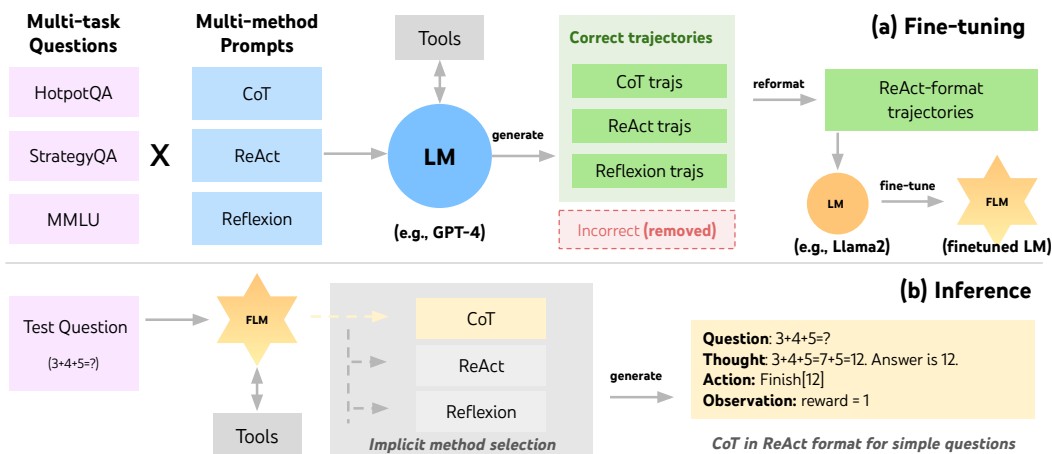

Figure 2: Illustration of FireAct. (a) **During fine-tuning**, a large LM (e.g., GPT-4) generates task-solving trajectories based on questions from different datasets and prompts from different methods. The successful trajectories are then converted into the ReAct format to fine-tune a smaller LM. (b) **During inference**, the fine-tuned LM could operate without few-shot prompting, and could implicitly select an prompting method to complete a ReAct trajectory with flexible lengths, adapting to different question complexities. For example, a simple question could be solved using only one thought-action-observation round, without using tools.

2023), human (Zhang et al., 2020), and multi-agent (Park et al., 2023) interactions. We refer readers to Xi et al. (2023) for an empirical survey and Sumers et al. (2023) for a systematic theoretical framework of language agents. Notably, most existing language agents prompted off-the-shelf LMs.

**Language model fine-tuning.** Adapting pre-trained LMs to downstream tasks is another active field of study (Zhang et al., 2023b), including various instruction-based fine-tuning datasets (Mishra et al., 2022; Sanh et al., 2022; Köpf et al., 2023; Wang et al., 2023d; Honovich et al., 2023; Longpre et al., 2023), models (Taori et al., 2023; Chiang et al., 2023; Xu et al., 2023; Muennighoff et al., 2023; Ouyang et al., 2022), parameter-efficient fine-tuning methods (Hu et al., 2022; Ding et al., 2023; Lv et al., 2023; Dettmers et al., 2023; Ivison et al., 2023), and data selection principles (Zhou et al., 2023a; Gunasekar et al., 2023). Additionally, there are various studies on fine-tuning specific types of LMs, such as coding LMs (Li et al., 2023; Luo et al., 2023; Rozière et al., 2023), multi-modal LMs (Zhang et al., 2023c; Gong et al., 2023; Dai et al., 2023; Zhang et al., 2023a; Brooks et al., 2023; Su et al., 2023), and retrieval-augmented LMs (Guu et al., 2020; Wang et al., 2023c). However, fine-tuning LMs for language agents that reason and act has been limited.

**Language agent fine-tuning.** Despite the vast interests in language agents and fine-tuning, their intersection has received limited attention, with only some initial study about how performances scale with the model size for a particular model family (Nakano et al., 2021; Schick et al., 2023; Yao et al., 2022b), how to incorporate more tools via retrieval (Patil et al., 2023; Qin et al., 2023), and some task-specific ablations (Yao et al., 2022a; Le et al., 2022). This paper takes on a more systematic investigation, proposing and answering new questions toward language agent fine-tuning.

## 3 FIREACT: FINE-TUNING LMS WITH DIVERSE REACT TRAJECTORIES

Our work is largely based on ReAct (Yao et al., 2022b), a popular approach to language agents. A ReAct task-solving trajectory (Figure 5) consists of multiple thought-action-observation rounds, where an LM generates free-form "thoughts" for versatile purposes (e.g., extract information from observations, propose and adjust action plans, track task progress), and structured "actions" to interact with environments (tools) and receive "observation" feedback. ReAct outperforms reasoning or acting only baselines, as reasoning can guide acting, and acting can support reasoning with new information. The ReAct format has thus been a basis of many follow-up language agents, such as Reflexion (Shinn et al., 2023), SwiftSage (Lin et al., 2023), and AutoGPT (Richards, 2023).

Also shown in (Yao et al., 2022b) was a preliminary PaLM (Chowdhery et al., 2022) fine-tuning experiment on HotpotQA (Yang et al., 2018), where a fine-tuned PaLM-62B outperforms a prompted PaLM-540B. But it remains unknown if such a finding generalizes to other types of LMs, prompting methods, or tasks. Follow-up studies on language agent fine-tuning have been sparse (see Section 2).

Thus we propose `FireAct`, a novel fine-tuning approach to language agents. As shown in Figure 2(a), `FireAct` also leverages few-shot prompting of a strong LM to generate diverse `ReAct` trajectories to fine-tune a smaller LM (i.e., distillation (Hinton et al., 2015)). But different from Yao et al. (2022b), `FireAct` explicitly promotes data diversity by mixing multiple training tasks and prompting methods. Here we consider two other methods compatible with the `ReAct` format:

- **Chain of Thought (CoT)** (Wei et al., 2022b) generates intermediate reasoning to bridge the question-answer gap. Each CoT trajectory can be turned into a simple one-round `ReAct` trajectory, with "thought" being the intermediate reasoning and "action" being returning the answer. CoT is useful for simple questions without tool needs (Figure 2(b)).

- **Reflexion** (Shinn et al., 2023) mostly follows the `ReAct` trajectory, but incorporates extra feedback and self-reflections. In this work, we simply prompt for reflections at the 6th and 10th `ReAct` round, so that long `ReAct` trajectories could pivot the strategy for solving the current task (e.g., "film search has not been helpful yet, I should search directors now").

During inference (Figure 2(b)), a `FireAct` agent alleviates the need for few-shot prompting, which makes inference more efficient and convenient. It could also implicitly select the suitable method adaptive to the task complexity, and show stronger generalization and robustness than prompting as a result of a wider and more diverse learning support.

## 4 EXPERIMENTAL SETUP

**Tasks**. Following prior work (Wei et al., 2022b; Yao et al., 2022b; Shinn et al., 2023), we train and test on well-established question answering (QA) tasks, which enjoy abundant and high-quality training data plus easy and faithful evaluation (answer exact match). We use four datasets:

- **HotpotQA** (Yang et al., 2018) is a QA dataset challenging multi-step reasoning and knowledge retrieval. The answer is usually a short entity or yes/no. We use 2,000 random training questions for fine-tuning data curation, and 500 random dev questions for evaluation.

- **Bamboogle** (Press et al., 2022) is a test set of 125 multi-hop questions with similar formats as HotpotQA, but carefully crafted to avoid direct solving with Google search.

- **StrategyQA** (Geva et al., 2021) is a yes/no QA dataset requiring implicit reasoning steps.

- **MMLU** (Hendrycks et al., 2021) covers 57 multi-choice QA tasks in various domains such as elementary mathematics, history, and computer science.

**Tool**. Following Press et al. (2022), we use SerpAPI[1] to build a Google search tool that returns the first existent item from "answer box", "answer snippet", "highlight words", or "first result snippet", which ensures the response is short and relevant. We find such a simple tool sufficient for basic QA needs across tasks, and increases our fine-tuned models' ease of use and generality.

**LMs**. We investigate three families of LMs:

- **OpenAI GPT**. We prompt GPT-4 (OpenAI, 2023b) to generate all fine-tuning data, and use GPT-3.5 for fine-tuning (OpenAI, 2023a) as well as prompting. We used both models in ChatCompletion mode from July to Sept 2023.

- **Llama-2** (Touvron et al., 2023b) with 7B and 13B parameters in "chat" mode.

- **CodeLlama** (Rozière et al., 2023) with 7B, 13B, and 34B parameters in "instruct" mode, which help further understand model size scaling and the importance of code fine-tuning for agentic tasks.

---

[1]`https://serpapi.com`.

**Fine-tuning methods**. We use Low-Rank Adaptation (LoRA) (Hu et al., 2022) for most fine-tuning experiments, but also use full-model fine-tuning for some comparison.

Given the various factors underlying language agent fine-tuning, We split experiments into three parts with increasing complexities:

- Fine-tuning using a single prompting method on a single task (Section 5);
- Fine-tuning using multiple methods on a single task (Section 6);
- Fine-tuning using multiple methods on multiple tasks (Section 7).

## 5   SINGLE-TASK, SINGLE-METHOD FINE-TUNING

In this section, we focus on fine-tuning with data from a single task (HotpotQA) and a single prompting method (`ReAct`). Using such a simple and controlled setup, we confirm various benefits of fine-tuning over prompting (performance, efficiency, robustness, generalization), and study effects of different LMs, data sizes, and fine-tuning methods. By default, we use 500 successful few-shot prompting trajectories generated by GPT-4 for training and a random subset of 500 HotpotQA dev questions for evaluation. Other experimental details can be found in the Appendix B.

### 5.1   PERFORMANCE AND EFFICIENCY

Table 1: Prompting results.   Table 2: Prompting vs. fine-tuning, with absolute/relative increases.

| | Prompt | EM |
|---|---|---|
| | IO | 37.2 |
| GPT-4 | CoT | **45.0** |
| | ReAct | 42.0 |
| | IO | 22.4 |
| GPT-3.5 | CoT | 28.0 |
| | ReAct | **31.4** |

| | ReAct | FireAct | abs./rel. diff |
|---|---|---|---|
| Llama-2-7B | 14.8 | 26.2 | +11.4 / **77%** |
| Llama-2-13B | 21.2 | 34.4 | **+13.1** / 62% |
| CodeLlama-7B | 17.4 | 27.8 | +10.4 / 60% |
| CodeLlama-13B | 20.8 | 29.0 | +8.2 / 39% |
| CodeLlama-34B | 22.2 | 27.8 | +5.6 / 25% |
| GPT-3.5 | **31.4** | **39.2** | +7.8 / 25% |

**Fine-tuning significantly increases agent performances**. As shown in Table 2, fine-tuning consistently and significantly improves the HotpotQA EM from prompting. While weaker LMs benefit more from fine-tuning (e.g., Llama-2-7B increases by 77%), even strong LMs such as GPT-3.5 could improve performances by 25%, clearly showing the benefit of learning from more samples. When compared to strong prompting baselines in Table 1, we find fine-tuned Llama-2-13B could outperform all GPT-3.5 prompting methods (Input-Output prompting, IO; Chain-of-thought, CoT; `ReAct`). It is a promising signal that fine-tuning small open-source LMs could outperform prompting stronger commercial LMs. Finally, fine-tuned GPT-3.5, which is the strongest fine-tuned LM, could outperform GPT-4 + IO prompting but still lags behind GPT-4 + CoT/`ReAct` prompting, suggesting room for improvement. More results (e.g., standard error) are in Appendix A.1.

**Fine-tuning is cheaper and faster during agent inference**. Since few-shot in-context examples are not needed for fine-tuned LMs, their inference becomes more efficient, especially for agentic applications where the context is iteratively accumulated. For example, the first part of Table 3 compares costs of fine-tuned vs. prompted GPT-3.5 inference, and finds the inference time is reduced by 70% (9.0s to 2.7s per trial), and the inference cost is reduced even though fine-tuned inference is charged $8\times$ expensive. While these costs will vary by conditions (e.g., parallelism implementation), the advantage of having a much smaller context is clear.

### 5.2   ROBUSTNESS AND GENERALIZATION

**Robustness to noisy tools**. The tools or environments that language agents interact with are not always trustworthy, which has led to safety concerns like jailbreaking (Liu et al., 2023b) or prompt injection (Willison, 2023). Here we consider a simplified and harmless setup, where the search API has a probability of 0.5 to return 1) "None" or 2) a random search response (from all previous experiments and trials), and ask if language agents could still robustly answer questions. As shown

Table 3: Comparison of costs, robustness, and generalization for fine-tuned vs. prompted GPT-3.5.

| | Cost per trial | | Obs. Robustness (EM) | | | Generalization |
| | Money ($) | Time (s) | Normal | "None" | Random | Bamboogle (EM) |
|---|---|---|---|---|---|---|
| FireAct | $\mathbf{2.2 \times 10^{-3}}$ | **2.7** | **39.2** | **33.6** | **37.2** | **44.0** |
| ReAct | $2.6 \times 10^{-3}$ | 9.0 | 31.4 | 20.8 | 22.6 | 40.8 |

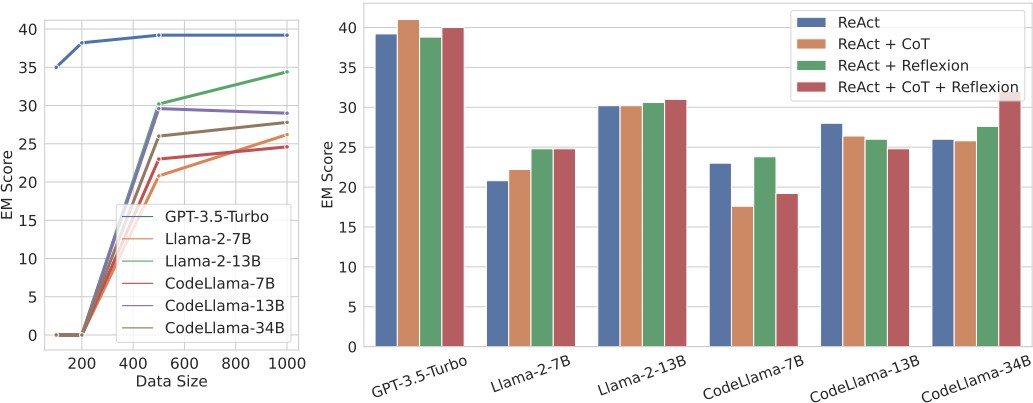

Figure 3: Data scaling.

Figure 4: Results across different LMs and data types.

in the second part of Table 3, the "None" setup turns out the more challenging one, which lowered `ReAct` EM by 33.8% and `FireAct` EM only by 14.2%. Interestingly, random observations hurt `ReAct` by a similar degree (28.0% drop), but does not hurt `FireAct` much (only 5.1% drop), possibly because the fine-tuning trajectories already contain examples of noisy search queries and how GPT-4 "reacts" to such noises successfully. These initial results hint at the importance of a more diverse learning support for robustness. More results on robustness can be found in Appendix A.2.

**Generalization to new tasks**. Table 3's third part shows EM results of fine-tuned and prompted GPT-3.5 on Bamboogle (Press et al., 2022), a test set of 125 multi-hop questions carefully crafted such that searching the questions on Google cannot directly give answers. While HotpotQA fine-tuned or prompted GPT-3.5 both generalize to Bamboogle reasonably, the former (44.0 EM) still beats the latter (40.8 EM), suggesting generalization advantages of fine-tuning. Similarly, combined with the few-shot prompts, fine-tuning on HotpotQA greatly improves the performance on Bamboogle, while slightly improving on MMLU and downgrading on StrategyQA compared to vanilla models (Appendix A.9). Since fine-tuning on HotpotQA could hardly generalize to StrategyQA (yes/no questions) or MMLU (multi-choice questions), two other QA datasets with different question styles and answer formats, it motivates our multi-task fine-tuning experiments in Section 7.

### 5.3 ANALYSIS OF VARIOUS FINE-TUNING FACTORS

**Effect of fine-tuning method (LoRA vs. Full model)**. For Llama-2-7B, we observe that full-model fine-tuning (30.2 EM) outperforms LoRA fine-tuning (26.2 EM) by 15.3% (see Appendix A.5). However, LoRA training is much more affordable, which can train 5.4 examples per second on a single RTX 4090 with 24GB GPU memory, while training 19.7 examples by full fine-tuning requires four A100 GPUs with 80GB GPU memory. Hence, running most experiments with LoRA allows us to explore more training settings with a limited budget and time frame.

**Effect of fine-tuning data scale**. Figure 3 shows how `FireAct` performances scale with the number of fine-tuning trajectories ($n \in \{100, 200, 500, 1000\}$). GPT-3.5 appears very sample-efficient, requiring only 100 samples to reach an EM around 35, and the gain after 200 samples is marginal. On the other hand, Llama models cannot even learn the `ReAct` format using 100 or 200 samples, but non-trivial scores "emerge" with 500 samples, and most models (except CodeLlama-13B) further improve with 1,000 samples. Such a data scaling trend suggests that smaller open-source LMs could

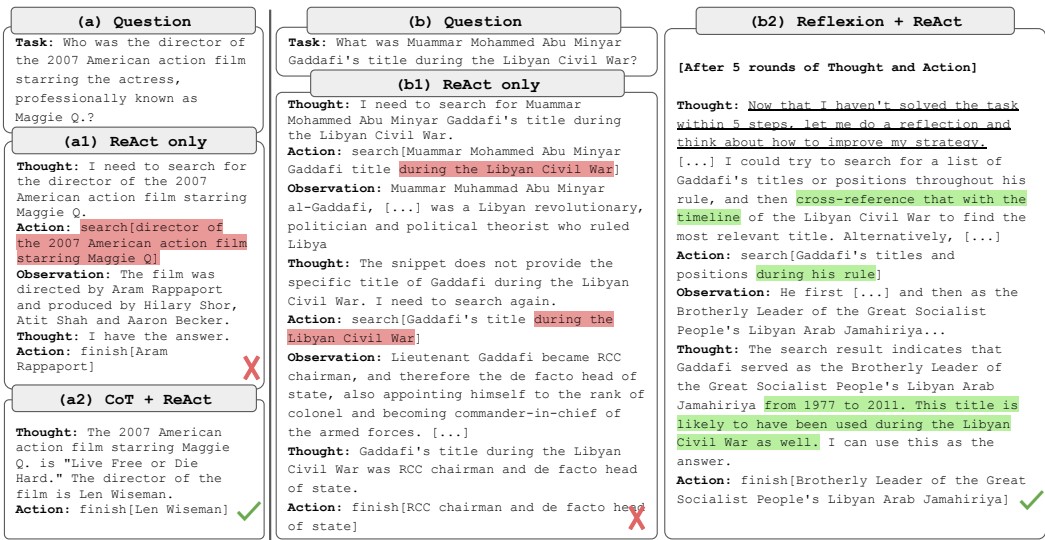

Figure 5: **For question (a)**, (a1) `ReAct`-only fine-tuning leads to distracting information and wrong answer, while (a2) CoT + `ReAct` fine-tuning leads to a simple CoT solution. **For question (b)**, (b1) `ReAct`-only fine-tuning leads to mutiple failed searches with the same patterns, while (b2) `Reflexion` + `ReAct` fine-tuning leads to a successful pivot of the search strategy.

potentially catch up with stronger LMs on a particular agentic task given enough fine-tuning data (e.g., Llama-2-13B fine-tuned on 1,000 samples can match GPT-3.5 fine-tuned on 100 samples).

**Effect of base LM type**. Table 2 reveals that GPT-3.5 is superior to all Llama-based models in both prompting and fine-tuning configurations. Additionally, CodeLlama-7B outperforms Llama-2-7B, while CodeLlama-13B does not perform as well as Llama-2-13B, suggesting that coding fine-tuning may not always be beneficial for agentic use cases. CodeLlama performs slightly better when using the default CodeLlama tokenizer instead of the Llama tokenizer (Appendix A.6).

**Effect of base LM scale**. As can be seen in Table 2 or the blue bars of Figure 4, (Code)Llama models with 13B parameters always outperform ones with 7B parameters, but CodeLlama-34B seems worse than CodeLlama-13B when fine-tuned purely on `ReAct` trajectories. But as we will see in Section 6 (and hinted in the rest of Figure 4), other factors such as the fine-tuning data type might affect the conclusion and make CodeLlama-34B outperforming CodeLlama-13B. In general, multiple components (LM type, LM scale, fine-tuning data and method) might influence fine-tuning results jointly, so different dimensions of scaling trends and LM/data types should also be considered jointly for agent design.

## 6 MULTI-METHOD FINE-TUNING

Next we integrate CoT (Wei et al., 2022b) and `Reflexion` (Shinn et al., 2023) with `ReAct` for multi-method fine-tuning on HotpotQA. For both methods, we generate 500 few-shot prompting trajectories via GPT-4, and use 47 long `Reflexion` trajectories that incorporated self-reflections after 6 or 10 `ReAct` rounds, and 187 successful CoT trajectories reformatted as single-round `ReAct` trajectories, on top of 500 existing `ReAct` trajectories. More details are in Appendix B.

**Multi-method fine-tuning increases agent flexibility**. Before quantitative results, we present two example questions in Figure 5 and some fine-tuned GPT-3.5 trajectories to illustrate the benefit of multi-method `FireAct` fine-tuning. The first question (a) is simple, but the `ReAct`-only fine-tuned agent (a1) searched an over-complicated query that led to the distraction and wrong answer. In contrast, an agent fine-tunend with both CoT and `ReAct` chose to solve the task within one round relying on confident internal knowledge. The second question (b) is harder, and the `ReAct`-only fine-tuned agent (b1) kept searching queries ending in "during the Libyan Civil War" without useful information. In contrast, an agent fine-tuned with both Reflexion and `ReAct` reflected upon this

Table 4: Multi-method results on HotpotQA.

| prompting method | EM | #Turns | |
|---|---|---|---|
| | | $\mu$ | $\sigma$ |
| ReAct | 39.4 | 3.2 | 1.4 |
| ReAct + CoT | **41.0** | 2.7 | 1.7 |
| ReAct + Reflexion | 38.8 | **3.8** | 2.8 |
| ReAct + CoT + Reflexion | 40.0 | 3.0 | 4.8 |
| Random method choice | 32.4 | - | - |
| Oracle method choice | 52.0 | - | - |

Table 5: Multi-task results with GPT-3.5.

| | HotpotQA | StrategyQA | Bamboogle | MMLU |
|---|---|---|---|---|
| **Prompting** | | | | |
| IO | 22.4 | 48.0 | 7.2 | **68.6** |
| CoT | 28.0 | 49.0 | 41.6 | 50.8 |
| ReAct | 31.4 | 61.0 | 40.8 | 58.6 |
| **Fine-tuning** | | | | |
| HotpotQA | 39.2 | - | 44.0 | - |
| Multi-task | 39.2 | 55.5 | 43.2 | 63.2 |
| + CoT | **39.6** | **72.9** | **50.4** | 65.8 |

problem, and pivoted the search strategy to change the time constraint to "during his rule", which led to the right answer. The flexibility to implicitly choose methods for different problems is another key advantage of fine-tuning over prompting.

**Multi-method fine-tuning affect different LMs differently**. Despite the intuitive benefit, Figure 4 shows mixing more methods does not always improve results, and the optimal mix of methods depends on the base LM. For example, ReAct+CoT outperforms ReAct for GPT-3.5 and Llama-2 models, but hurts for CodeLlama models. ReAct+CoT+Reflexion is the worst for CodeLlama-7/13B, but is the best for CodeLlama-34B. These non-trivial results call for further studies of the interaction of base LMs and fine-tuning data.

**Can multi-method agents choose suitable methods?** Table 4 displays HotpotQA test results of various FireAct agents based on GPT-3.5, as well as the mean ($\mu$) and standard deviation ($\sigma$) of the number of ReAct rounds across their trajectories. Compared to ReAct-only fine-tuning, ReAct+CoT improves the EM and reduces the trajectory length, while ReAct+Reflexion hurts the EM and increases the trajectory length. This suggests the two method mixes shift the method selection to two different directions, and CoT is perhaps more helpful for HotpotQA questions. To further understand if multi-method agents could choose the suitable methods, we calculate the result of randomly choosing a method during inference. The result of 32.4 is much lower than all multi-method agents, suggesting the method selection is non-trivial. But applying the best method for each instance leads to an "oracle" result of 52.0, suggesting room for improving prompting method selection. Future work could explore more systematic grid search or connections between trajectory statistics and performances to set up better method mix ratios.

## 7    MULTI-TASK FINE-TUNING

So far fine-tuning has only used HotpotQA data, but empirical studies on LM fine-tuning have shown the benefit of mixing different tasks (Longpre et al., 2023). Here we fine-tune GPT-3.5 using a mix of training data from three datasets: HotpotQA (500 ReAct samples, 277 CoT samples), StrategyQA (388 ReAct samples, 380 CoT samples), and MMLU (456 ReAct samples, 469 CoT samples). These samples are picked from successful ReAct/CoT few-shot prompting trajectories generated via GPT-4.

As shown in Table 5, when StrategyQA/MMLU data is added ("Multi-task"), HotpotQA/Bamboogle performances almost remain the same. On one hand, StrategyQA/MMLU trajectories contain very different questions (e.g., MMLU questions are multi-choice) and tool use strategies (e.g., MMLU ReAct trajectories tend to search answer choices), which makes transfer hard. On the other hand, despite the distribution shift, adding StrategyQA/MMLU does not hurt HotpotQA/Bamboogle performances, which hints at the promise of fine-tuning one multi-task agent to replace multiple single-task agents, capturing the performance improvement of fine-tuning based agents, without sacrificing the flexibility of prompting based agents or worrying about negative cross-task influences.

When we switch from multi-task, single-method fine-tuning to multi-task, multi-method fine-tuning, we find increased performances across all tasks, again reinforcing the value of multi-method agent fine-tuning. Intriguingly, all fine-tuned agents (plus CoT/ReAct prompting) underperform naive input-output (IO) prompting on MMLU. One possible explanation is these questions might be too easy to require reasoning and acting, and another explanation could be answer choice memorization. This urges efforts for better prompting methods as well as for better agent datasets.

## 8  DISCUSSION

**When to fine-tune vs. prompt for language agents?**  While most existing language agents use prompting, our work calls for a re-thinking of best practices by showing multi-facet benefits of fine-tuning as a result of more diverse learning support. Thus, prompting and fine-tuning seem more suitable for exploration and exploitation usecases respectively. To develop new agents or solve new tasks, prompting off-the-shelf LMs could provide flexibility and convenience. On the other hand, when the downstream task is known (e.g., QA), effective prompting methods for agents have been explored (e.g., ReAct), and enough data can be collected (e.g., via GPT-4), fine-tuning can provide better performances, stronger generalization to new tasks, more robustness to noisy or adversarial environments, as well as cheaper and more efficient inference. These features make fine-tuning especially attractive when used for large-scale industrial solutions.

**Which LM to fine-tune?**  Of all models we considered, GPT-3.5 consistently outperforms other Llama-based LMs in various setups, which is not surprising given its much larger model size and continued training from GPT-3. It also has better sample efficiency and a reasonable cost (around $10 per fine-tuning experiment in our case). However, we have also shown that open source Llama models could be fine-tuned to catch up with GPT-3.5 performances, given enough fine-tuning data with the right mixing of prompting methods and tasks. Practioners should balance the tradeoff between the convenience and performance of GPT-3.5 versus the controlability and reproducibility of open-source LMs for agent fine-tuning.

**When to use tools or reflect for language agents?**  Prompting-based language agents can only imitate a small and fixed set of successful task-solving trajectories. This could lead to tool overuse (e.g., search for knowledge already stored in LMs), and inabilities to recover when the trajectory deviates from the "successful" patterns (e.g., keep searching similar queries with useless observations). `FireAct`'s multi-method fine-tuning helps increase a language agent's flexibility and robustness, but the problem of knowing when to get help (tool use) and feedback (reflection) is still far from being solved. Work on calibration (Ren et al., 2023) and meta-reasoning (Griffiths et al., 2019) might shed light into better agent designs in this regard.

**Limitations and future directions**. This work is an initial step toward language agent fine-tuning, and is constrained to a single type of task (QA) and a single tool (Google search). Future work could apply the research questions raised by `FireAct` to more tasks and grounding setups (e.g., more API tools, the web, the physical world). Also, we focused on three methods (`ReAct`, `CoT`, `Reflexion`) that maintain a single autoregressive trajectory context, which makes fine-tuning straightforward. It remains underexplored how to fine-tune more advanced agents involving multiple prompts, roles, and contexts (Wang et al., 2023a; Park et al., 2023; Yao et al., 2023), model multi-agent interaction and orchestration, or best combine prompting and fine-tuning in a complex agent system. These are exciting future directions for fine-tuning based language agents. Finally, the multi-task setup in this work is limited to three QA tasks, and the best LM we could fine-tune is GPT-3.5. A large-scale multi-task fine-tuning (Wei et al., 2022a) using the state-of-the-art LM backbone will test the limit of language agent fine-tuning, but more suitable and diverse benchmarks to develop and evaluate agents should be explored first.

REPRODUCIBILITY STATEMENT

Our main experiments are performed on API-based GPT4 [2] and GPT-3.5-Turbo[3], and open source Llama[4] and CodeLlama[5]. Details of the experiment setting are in Appendix B and all used prompts are in Appendix C. The codebase is released at: `https://anonymous.4open.science/r/FireAct-DC39/`.

---

[2]`https://openai.com/research/gpt-4`
[3]`https://openai.com/blog/gpt-3-5-turbo-fine-tuning-and-api-updates`
[4]`https://huggingface.co/meta-llama`
[5]`https://huggingface.co/docs/transformers/main/model_doc/code_llama`

ETHICS STATEMENT

This research focuses on language agents, and we are aware of the potential risks associated with uncontrolled autonomous interactions. Thus, we have chosen a setup of open-domain question answering with access to a Google search API, where the API is read-only and does not cause any changes to the Internet. For the robustness study, we change the Google search API responses to empty strings or random Google responses, and does not cause the agent to receive malicious or hateful observations. Our investigations on the generalization and robustness of language agents will contribute to their safe deployment.

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

# A  ADDITIONAL RESULTS

## A.1  STANDARD ERROR OF EM SCORES

Table 6: Prompting results with Standard Errors ($\sigma_M$).

|  | Prompt | EM | $\sigma_M$ |
|---|---|---|---|
|  | IO | 37.2 | 2.16 |
| GPT-4 | CoT | **45.0** | 2.22 |
|  | ReAct | 42.0 | 2.21 |
|  | IO | 22.4 | 1.86 |
| GPT-3.5 | CoT | 28.0 | 2.01 |
|  | ReAct | **31.4** | 2.08 |

Table 7: Prompting vs. fine-tuning, with absolute/relative increases and Standard Errors ($\sigma_M$).

|  | ReAct | FireAct | abs./rel. diff | $\sigma_{ReAct}$ | $\sigma_{FireAct}$ |
|---|---|---|---|---|---|
| Llama-2-7B | 14.8 | 26.2 | +11.4 / **77%** | 1.59 | 1.97 |
| Llama-2-13B | 21.2 | 34.4 | **+13.1** / 62% | 1.83 | 2.12 |
| CodeLlama-7B | 17.4 | 27.8 | +10.4 / 60% | 1.69 | 2.00 |
| CodeLlama-13B | 20.8 | 29.0 | +8.2 / 39% | 1.82 | 2.03 |
| CodeLlama-34B | 22.2 | 27.8 | +5.6 / 25% | 1.86 | 2.00 |
| GPT-3.5 | **31.4** | **39.2** | +7.8 / 25% | 2.08 | 2.18 |

Table 8: Multi-task results with GPT-3.5 and Standard Errors ($\sigma_M$).

|  | HotpotQA | | | | StrategyQA | | | |
|---|---|---|---|---|---|---|---|---|
|  | EM | $\sigma_M$ | EM | $\sigma_M$ | EM | $\sigma_M$ | EM | $\sigma_M$ |
| **Prompting** | | | | | | | | |
| IO | 22.4 | 1.86 | 48.0 | 2.23 | 7.2 | 1.16 | 68.6 | 2.08 |
| CoT | 28.0 | 2.01 | 49.0 | 2.24 | 41.6 | 2.20 | 50.8 | 2.24 |
| ReAct | 31.4 | 2.08 | 61.0 | 2.18 | 40.8 | 2.20 | 58.6 | 2.20 |
| **Fine-tuning** | | | | | | | | |
| HotpotQA | 39.2 | 2.18 | - | - | 44.0 | 2.22 | - | - |
| Multi-task | 39.2 | 2.18 | 55.5 | 2.19 | 43.2 | 2.20 | 63.2 | 2.16 |
| + CoT | **39.6** | 2.19 | **72.9** | 1.99 | **50.4** | 2.24 | 65.8 | 2.12 |

## A.2  ROBUSTNESS ANALYSIS

We append more results on robustness analysis in Table 9.

## A.3  DATA MIX

The Table 10 illustrates the lower and upper performance limits of the model. The lower boundary is determined by randomly selecting one of the agent methods, while the upper boundary is established by always selecting the best agent method when a question is asked. The lower and upper bounds yielded 32.4 and 52.0 (EM), respectively. The goal of the language agent fine-tuning with mixed agent methods is to reach the theoretical optimum, which is impractical but optimal. The results of the data mix demonstrate that the performance of all data mix falls within the range of the mean and the best baselines, indicating a consistent improvement. Notably, the inclusion of the CoT method is especially advantageous, resulting in a significant improvement in performance. Despite the relatively poor performance of CoT individually (28.0 EM as suggested in Table 1), their combined data contribute positively to the fine-tuning process. Table 10 also summarizes our research into the effects of mixed agent methods applied to fine-tuned language agents. The results are varied, with the EM performance of the agent, when various mixed methods are used, ranging from 38.8 to 41.0.

|  |  | Normal | None | Random |
|---|---|---|---|---|
| ReAct |  | 31.4 | 20.8 | 22.6 |
| FireAct | Single Task Single Method | 39.2 | 33.6 | 37.2 |
|  | Multi-task, Single Method | 39.2 | 34.8 | 36.2 |
|  | Single Task, Multi-method | 41.0 | 36.4 | 38.4 |
|  | Multi-task, Multi-method | 39.6 | 35.2 | 37.0 |

Table 9: Comparison of results for different tasks and methods.

| Multi-methods Language Agent Finetuning |  | EM | Number of Turns $\mu$ | $\sigma$ |
|---|---|---|---|---|
| Single Agent Method | IO (Prompt) | 22.4 | 0 | 0 |
|  | CoT (Prompt) | 28.0 | 0.8 | 0.4 |
|  | ReAct (Finetune) | 39.4 | 3.2 | 1.4 |
|  | Reflexion (Finetune) | 39.8 | 3.5 | 2.6 |

| Chosen Agent Method |  | EM | Number of Turns $\mu$ | $\sigma$ |
|---|---|---|---|---|
| Practical | Randomly choose one | 32.4 | - | - |
|  | Best single method | 39.8 | 3.5 | 2.6 |
| Theoretically optimal | Always choose the best one | 52.0 | 1.3 | 1.1 |

| Mix agent methods FireAct | ReAct | IO | CoT | Reflexion | EM | Number of Turns $\mu$ | $\sigma$ |
|---|---|---|---|---|---|---|---|
|  | ✓ | ✓ | ✓ | ✓ | 39.4 | 2.4 | 1.8 |
|  | ✓ | ✓ | ✓ | ✗ | 41.0 | 2.6 | 1.6 |
|  | ✓ | ✓ | ✗ | ✓ | 41.2 | 3.3 | 1.6 |
|  | ✓ | ✓ | ✗ | ✗ | 40.2 | 3.3 | 1.5 |
|  | ✓ | ✗ | ✓ | ✓ | 40.0 | 3.0 | 4.8 |
|  | ✓ | ✗ | ✓ | ✗ | 41.0 | 2.7 | 1.7 |
|  | ✓ | ✗ | ✗ | ✓ | 38.8 | 3.8 | 2.8 |

Table 10: Mixed agent methods investigation on HotpotQA

## A.4 TURN DISTRIBUTION

The graph in Figure 6 demonstrates that combining data changes the distribution to resemble that of the training data. This was especially noticeable when contrasting ReAct+CoT (2.7 turns on average, 41.0 EM) and ReAct+CoT (3.8 turns on average, 38.8 EM).

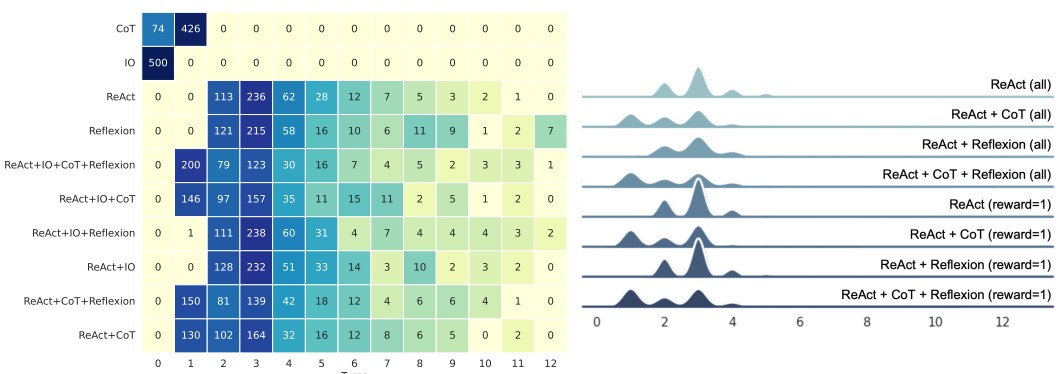

Figure 6: Turn distribution

## A.5 LoRA vs. Full Fine-tuning

|  | LoRA | Full |
|---|---|---|
| $EM_{\text{single}}$ | 26.2 | 30.2 |
| $EM_{\text{multi}}$ | 27.0 | 30.0 |
| $M_{\text{size}}$ | 38MB | 14GB |

Table 11: LoRA vs. Full fine-tuning on single-task setting and multi-task setting.

Table 11 demonstrates the EM score of LLama-2-7B fine-tuned on 500 HotpotQA trajectories (single-task setting) and mixed all trajectories (multi-task setting) with LoRA and full weights respectively.

## A.6 CodeLLama vs. Llama Tokenizer

|  | Llama Tokenizer | CodeLLama Tokenizer | abs./rel. diff |
|---|---|---|---|
| CodeLlama-7B | 26.6 | 27.8 | +1.2 / 4.5% |
| CodeLlama-13B | 27.4 | 29.0 | +1.6 / 5.8% |

Table 12: The performance difference of CodeLlama between using Llama tokenizers and CodeLlama tokenizers.

Table 12 compares the Llama Tokenizer and the CodeLLama Tokenizer in terms of their impact on the performance of the CodeLlama model. EM scores are provided for two variants of the CodeLlama model, CodeLlama-7B and CodeLlama-13B, for each tokenizer on HotpotQA.

## A.7 World modelling: masking observations

Table 13 suggests that the incorporation of observation masking generally leads to slight improvements in performance for the CodeLlama-7B model. The most significant improvement is seen in the ReAct + CoT setting. However, the influence of observation masking on the CodeLlama-13B model is not consistent across configurations, with some cases showing improvement and others not. This table provides a detailed overview of the effect of observation masking on the CodeLlama models' performance in HotpotQA tasks, with the performance metric values for each configuration presented for further study. It appears that in our current settings, learning world modelling does not have a consistent effect on performance.

|  | w/ Observation Mask | w/o Observation Mask |
|---|---|---|
| *CodeLlama-7B* |  |  |
| ReAct | 26.0 | **26.2** |
| ReAct + CoT | **24.0** | 22.2 |
| ReAct + Reflexion | **25.2** | 24.8 |
| ReAct + Reflexion + CoT | **26.0** | 24.8 |
| *CodeLlama-13B* |  |  |
| ReAct | **34.6** | 34.4 |
| ReAct + CoT | 24.8 | **30.2** |
| ReAct + Reflexion | **30.6** | **30.6** |
| ReAct + Reflexion + CoT | 27.2 | **31.0** |

Table 13: The performance difference of fine-tuning Llama-2-7B and Llama-2-13B with or without masking the observations in HotpotQA trajectories.

## A.8 Training Epochs

Table 7 presents the performance changes of the GPT-3.5 model on the HotpotQA dataset as the number of fine-tuning epochs varies. As the number of fine-tuning epochs increases from 1 to 4,

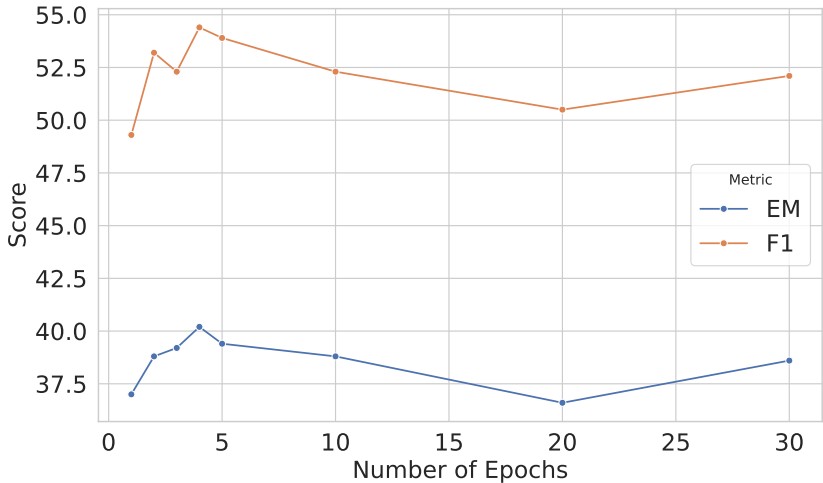

Figure 7: Performance changes of GPT-3.5-Turbo on HotpotQA dataset with increasing fine-tuning epochs

both the EM and F1 scores generally improve, indicating increased precision in providing exact answers and overall answer quality. However, beyond 4 epochs, while EM scores continue to increase slightly, F1 scores plateau and even dip at times, suggesting a diminishing return on additional fine-tuning.

## A.9 FEW-SHOT TASK GENERALIZATION OF FIREACT

|  | StrategyQA | Bamboogle | MMLU |
|---|---|---|---|
| *Fine-tuned Llama-2-7B + Few-shot Prompt* | | | |
| IO | 44.1 | 0 | 5.4 |
| CoT | 27.0 | 0 | 4.2 |
| ReAct | 52.0 | 35.2 | 34.0 |
| *Vanilla Models + Few-shot ReAct* | | | |
| Llama-2-7B | 59.0 | 11.2 | 33.2 |
| GPT-3.5 | 61.0 | 40.8 | 58.6 |

Table 14: Multi-task results with Llama-2-7B fine-tuned on HotpotQA trajectories.

## B EXPERIMENT DETAILS

### B.1 BASE MODELS

We explored the base models from three representative LLM families: (1) Generative Pre-trained Transformer (GPT) family that includes GPT-4 (OpenAI, 2023b) and GPT-3.5, Llama-2 (Touvron et al., 2023b), and CodeLlama (Rozière et al., 2023). GPT-3.5 is the instructed and reinforcement learning from human feedback (RLHF) version of GPT-3 (Brown et al., 2020) with 175B parameters and is capable of understanding and generating human-like text in a wide range of tasks and domains. GPT-4 is widely regarded as one of the state-of-the-art LLMs, and both GPT-3.5 and GPT-4 have the ability to use a sequence of tools in a zero-shot context with the feature of the plugin [6]. (2) LLama-2 is a series of open source pre-trained and fine-tuned LLMs ranging from 7B to 70B, which has preliminary evidence of tool use emergence, but its performance has yet to be extensively tested. (3) CodeLlama is a family of LLMs specifically designed for code generation derived from Llama-2 and ranging from 7B to 34B, which gradually specialises and increases the capabilities of Llama-2 models in coding by applying a cascade of training and fine-tuning steps.

---
[6]https://openai.com/blog/chatgpt-plugins

## B.2 Single-method Single-task Setup

We initially took samples from the HotpotQA Train set and asked GPT-4 models (OpenAI, 2023b) to generate 500 ReAct trajectories with few-shot examples for agent fine-tuning with human-in-the-loop validation. We also selected 500 examples from the original 7,405 Dev set for evaluation with the exact match (EM) and the F1 score as two metrics. GPT-3.5 was fine-tuned `GPT-3.5-Turbo-0613` with OpenAI fine-tuning API [7], while Llama-2 and CodeLlama were fine-tuned `Llama-2-Chat-HF` and `CodeLlama-Instruct-HF` on A100-80GB, RTX 6000 Ada, RTX 4090 and RTX 3090 GPUs with Low-Rank Adaptation (LoRA) (Hu et al., 2022) with `int8` quantization. We use OpenAI fine-tuning API (OpenAI, 2023a) to fine-tune `GPT-3.5-Turbo-0613` for 3 epochs and Low-Rank Adaptation (LoRA) (Hu et al., 2022) to fine-tune Llama-2 and CodeLlama families for 30 epochs. We set a hard limit if the `ReAct` format algorithms did not finish in 11 steps. The learning rate is `3e-4`, the trainning batch size is `16`. The evaluation temperature of the Llama and Codellama models is `0.6`, and the temperature of GPT-3.5 is `0`. We also evaluate the models also with `int8` quantization.

## B.3 Computation of Llama and CodeLlama

Table 15 demonstrates the computational resources used for fine-tuning and inferring Llama-2 and CodeLlama models.

|  | Training GPU | Example / Second | Inference GPU |
|---|---|---|---|
| *LoRA Fine-tuning* | | | |
| Llama-2-7B | 1 × RTX 4090 - 24GB | 5.4 | 1 × RTX 4090 - 24GB |
| Llama-2-13B | 1 × RTX 4090 - 24GB | 2.5 | 1 × RTX 4090 - 24GB |
| CodeLlama-7B | 1 × RTX 4090 - 24GB | 5.5 | 1 × RTX 4090 - 24GB |
| CodeLlama-13B | 1 × RTX 4090 - 24GB | 2.2 | 1 × RTX 4090 - 24GB |
| CodeLlama-34B | 1 × RTX 6000 Ada - 48GB | 0.9 | 1 × RTX 6000 Ada - 48GB |
| *Full Fine-tuning* | | | |
| Llama-2-7B | 4 × A100 - 80GB | 19.7 | 1 × RTX 4090 - 24GB |

Table 15: The computational resources used for training and inference with Llama-2 and CodeLlama models.

---

[7] `https://platform.openai.com/docs/guides/fine-tuning`

## C  PROMPTS

Hotpot QA Prompts

---

**Input-Output (IO)**

| | |
|---|---|
| Question | What is the elevation range for the area that the eastern sector of the Colorado orogeny extends into? |
| Answer | 1,800 to 7,000 ft |
| | |
| Question | Musician and satirist Allie Goertz wrote a song about the "The Simpsons" character Milhouse, who Matt Groening named after who? |
| Answer | Richard Nixon |
| | |
| Question | Which documentary is about Finnish rock groups, Adam Clayton Powell or The Saimaa Gesture? |
| Answer | The Saimaa Gesture |
| | |
| Question | What profession does Nicholas Ray and Elia Kazan have in common? |
| Answer | director, screenwriter, actor |
| | |
| Question | Which magazine was started first Arthur's Magazine or First for Women? |
| Answer | Arthur's Magazine |
| | |
| Question | Were Pavel Urysohn and Leonid Levin known for the same type of work? |
| Answer | Yes |

---

**Chain-of-Thoughts (CoT)**

| | |
|---|---|
| Question | What is the elevation range for the area that the eastern sector of the Colorado orogeny extends into? |
| Thought | The eastern sector of Colorado orogeny extends into the High Plains.  High Plains rise in elevation from around 1,800 to 7,000 ft, so the answer is 1,800 to 7,000 ft. |
| Answer | 1,800 to 7,000 ft |
| | |
| Question | Musician and satirist Allie Goertz wrote a song about the "The Simpsons" character Milhouse, who Matt Groening named after who? |
| Thought | Milhouse was named after U.S. president Richard Nixon, so the answer is Richard Nixon. |
| Answer | Richard Nixon |
| | |
| Question | Which documentary is about Finnish rock groups, Adam Clayton Powell or The Saimaa Gesture? |
| Thought | Adam Clayton Powell (film) is a documentary about an African-American politician, not Finnish rock groups.  So the documentary about Finnish rock groups must instead be The Saimaa Gesture. |
| Answer | The Saimaa Gesture |
| | |
| Question | What profession does Nicholas Ray and Elia Kazan have in common? |
| Thought | Professions of Nicholas Ray are director, screenwriter, and actor. Professions of Elia Kazan are director, producer, screenwriter, and actor. So profession Nicholas Ray and Elia Kazan have in common is director, screenwriter, and actor. |
| Answer | director, screenwriter, actor |
| | |
| Question | Which magazine was started first Arthur's Magazine or First for Women? |
| Thought | Arthur's Magazine was started in 1844.  First for Women was started in 1989. 1844 (Arthur's Magazine) < 1989 (First for Women), so Arthur's Magazine was started first. |
| Answer | Arthur's Magazine |
| | |
| Question | Were Pavel Urysohn and Leonid Levin known for the same type of work? |
| Thought | Pavel Urysohn is a mathematician.  Leonid Levin is a mathematician and computer scientist.  So Pavel Urysohn and Leonid Levin have the same type of work. |
| Answer | Yes |

---

**ReAct**

| | |
|---|---|
| | Solve a question answering task with interleaving Thought, Action, Observation steps.  Thought can reason about the current situation, and Action can be two types: 
(1) search[question], which searches a question on Google and returns a short snippet containing the answer.  Note that sometimes the snippet does not contain the answer, and some alternative search might be needed. 
(2) finish[answer], which returns the answer and finishes the task. 
Here are some examples. |
| | |
| Question | What is the elevation range for the area that the eastern sector of the Colorado orogeny extends into? |

Hotpot QA Prompts – Continued from previous page

| | |
|---|---|
| Thought | I need to first find the eastern sector of the Colorado orogeny extends into what, then find its elevation range. |
| Action | search[the eastern sector of the Colorado orogeny extends into what?] |
| Observation 1 | the High Plains |
| Thought | I need to find the elevation range for the High Plains. |
| Action | search[elevation range of the High Plains?] |
| Observation | around 1,800 to 7,000 ft |
| Thought | I have the answer. |
| Action | finish[1,800 to 7,000 ft] |
| Observation | Episode finished, reward = 1 |
| | |
| Question | Musician and satirist Allie Goertz wrote a song about the "The Simpsons" character Milhouse, who Matt Groening named after who? |
| Thought | I need to search Milhouse is named after who. |
| Observation | U.S. president Richard Nixon |
| Thought | I find the answer. |
| Action | finish[Richard Nixon] |
| Observation | Episode finished, reward = 1 |
| | |
| Question | Which documentary is about Finnish rock groups, Adam Clayton Powell or The Saimaa Gesture? |
| Thought | I need to search documentary Adam Clayton Powell and documentary The Saimaa Gesture to find which is about Finnish rock groups. |
| Action | search[documentary Adam Clayton Powell] |
| Observation | Adam Clayton Powell (1989). Documentary. The Academy Award¨-nominated Adam Clayton Powell delves into the gripping life and career of the most influential ... |
| Thought | I do not get whether it is about Finnish rock groups. I need to search Adam Clayton Powell to make sure. |
| Action | search[Adam Clayton Powell] |
| Observation | Re-elected for nearly three decades, Powell became a powerful national politician of the Democratic Party, and served as a national spokesman on civil rights ... |
| Thought | Adam Clayton Powell is a politican, not Finnish rock groups. I need to search The Saimaa Gesture to make sure. |
| Action | search[The Saimaa Gesture documentary] |
| Observation | It is a documentary about three Finnish rock groups aboard the steamboat SS Heinävesi on their tour around Lake Saimaa. The Saimaa Gesture. Directed by, Aki ... |
| Thought | The Saimaa Gesture is about three Finnish rock groups, so the answer is The Saimaa Gesture. |
| Action | finish[The Saimaa Gesture] |
| Observation | Episode finished, reward = 1 |
| | |
| Question | What profession does Nicholas Ray and Elia Kazan have in common? |
| Thought | I need to search the profession of Nicholas Ray and Elia Kazan, then find what is common. |
| Action | search[Nicholas Ray profession] |
| Observation | New York City, U.S.. Occupation(s), Film director, screenwriter, actor. Years active, 1946{1979. Spouses. |
| Thought | Nicholas Ray is film director, screenwriter, actor. I need to search Elia Kazan next. |
| Action | search[Elia Kazan profession] |
| Observation | Occupations. Actor; director; producer; screenwriter. Years active, 1934 – 1976. Spouses. Molly Day Thacher Kazan... (m. 1932, until her death in 1963). |
| Thought | Elia Kazan is actor, director, producer, screenwriter. So the common profession is actor, director, screenwriter |
| Action | finish[actor, director, screenwriter] |
| Observation | Episode finished, reward = 1 |

MMLU Prompts

Input-Output(IO)

| | |
|---|---|
| Question | Single choice :  A person takes buckets of water from the house and begins to add it to a pond in the yard.  After a certain point, the pond
A. bloats
B. breaks
C. sinks
D. drowns |
| Answer | A |
| Question | Single choice :  Coal is solid rock that began as organic material that was deposited in a swamp.  The formation of coal suggests that,
A. coal is made mostly of skeletal remains of animals.
B. coal is formed from magma that has solidified over time.
C. it quickly becomes petrified when water is removed.
D. geologic processes continue over millions of years. |
| Answer | D |
| Question | Single choice :  A student uses the following characteristics to describe a group of objects in space.
* 200 billion stars
* 30 million light years from Earth
* 500 light years in diameter
Which of the following is the student most likely describing?
A. a galaxy
B. the universe
C. a constellation
D. the solar system |
| Answer | A |

Chain-of-Thoughts (CoT)

| | |
|---|---|
| Question | Single choice :  A person takes buckets of water from the house and begins to add it to a pond in the yard.  After a certain point, the pond
A. bloats
B. breaks
C. sinks
D. drowns |
| Thought | Each time the person adds a bucket of water, the level of water in the pond rises.  Of all options, only A. bloats is consistent with the rise of water level.  So the answer is A. |
| Answer | A |
| Question | Single choice:  Coal is solid rock that began as organic material that was deposited in a swamp.  The formation of coal suggests that,
A. coal is made mostly of skeletal remains of animals.
B. coal is formed from magma that has solidified over time.
C. it quickly becomes petrified when water is removed.
D. geologic processes continue over millions of years. |
| Thought | Let's evaluate each option.  A. Coal is mostly composed of plant matter, not the skeletal remains of animals.  Therefore, this option is incorrect. B. Coal is not formed from magma.  Magma that solidifies over time creates igneous rocks, so this option is also incorrect.  C. Petrification is a process by which organic material is turned into stone.  It is not directly related to the process of coal formation, so this option is incorrect.  D. The formation of coal takes incredibly long periods of time and consists of slow geologic processes such as sedimentation and metamorphism, which suggests that such processes continue over millions of years.  Therefore, this option is correct.  The answer is D. |
| Answer | D |
| Question | Single choice :  A student uses the following characteristics to describe a group of objects in space.
* 200 billion stars
* 30 million light years from Earth
* 500 light years in diameter
Which of the following is the student most likely describing?
A. a galaxy
B. the universe
C. a constellation
D. the solar system |
| Thought | Let's evaluate each option.  A. a galaxy :  Possibly, as galaxies do contain billions of stars and can be millions of light years from Earth.  B. the universe :  Unlikely, as the universe is far larger than 30 million light years and contains more than just 200 billion stars.  C. a constellation : Unlikely, as constellations are patterns of stars seen from Earth and don't have a physical size or distance associated with them.  D. the solar system :  Definitely not, as our solar system only contains one star, our sun.  So, the answer is most likely A. a galaxy. |
| Answer | A |

MMLU Prompts – Continued from previous page

**ReAct**

|  |  |
|---|---|
| | Solve a question answering task with interleaving Thought, Action, Observation steps. Thought can reason about the current situation, and Action can be two types:
(1) search[question], which searches a question on Google and returns a short snippet containing the answer. Note that sometimes the snippet does not contain the answer, and some alternative search might be needed.
(2) finish[answer], which returns the answer and finishes the task.
Here are some examples. |
| Question | Single choice : A person takes buckets of water from the house and begins to add it to a pond in the yard. After a certain point, the pond
A. bloats
B. breaks
C. sinks
D. drowns |
| Thought | After continuously adding water to a pond, the pond will have more water than it could hold, thus bloats. So the answer is A. |
| Action | finish[A] |
| Observation | Episode finished, reward = 1 |
| Question | Single choice : Coal is solid rock that began as organic material that was deposited in a swamp. The formation of coal suggests that,
A. coal is made mostly of skeletal remains of animals.
B. coal is formed from magma that has solidified over time.
C. it quickly becomes petrified when water is removed.
D. geologic processes continue over millions of years. |
| Thought | The question is about the formation of coal. I need to first learn how coal is formed. |
| Action | search[How is coal formed?] |
| Observation | Coal takes millions of years to form Coal contains the energy stored by plants that lived hundreds of millions of years ago in swampy forests. Layers of dirt and rock covered the plants over millions of years. The resulting pressure and heat turned the plants into the substance we call coal. |
| Thought | Based on the information, I can check each option. A : coal is made by plants, not animals, so A is false. B : I have no information about if coal is formed from magma yet. I could search "is coal formed from magma" to make sure. C : I have no information about if coal quickly becomes petrified when water is removed. I could search "does coal quicklybecome petrified when water is removed" to make sure. D : Coal takes millions of years to form, so D is possibly true. I could search "is the formulation of coal a geologic process" to make sure. |
| Action | search[is the formulation of coal a geologic process] |
| Observation | It is formed from plant remains that have been compacted, hardened, chemically altered, and metamorphosed by heat and pressure over geologic time. |
| Thought | Seems the formulation of coal is over geologic time, so a geologic process. So the answer is D. |
| Action | finish[D] |
| Observation | Episode finished, reward = 1 |
| Question | Single choice : A student uses the following characteristics to describe a group of objects in space.
* 200 billion stars
* 30 million light years from Earth
* 500 light years in diameter
Which of the following is the student most likely describing?
A. a galaxy
B. the universe
C. a constellation
D. the solar system |
| Thought | These options correspond to space systems of different sizes. I could search what is the diameter of each option to match. |
| Action | search[what is the diameter of a galaxy] |
| Observation | Most galaxies are 1,000 to 100,000 parsecs in diameter (approximately 3,000 to 300,000 light years) and are separated by distances on the order of millions of parsecs (or megaparsecs). |

**MMLU Prompts – Continued from previous page**

| | |
|---|---|
| Thought | A galaxy is usually 3,000 to 300,000 light years in diameter, which is slightly more than 500 light years. I should search the diameter of the universe next. |
| Action | search[what is the diameter of the universe] |
| Observation | 93 billion light-years |
| Thought | The universe is 93 billion light years in diameter, which is much larger than 500 light years. I should search the diameter of a constellation next. |
| Action | search[what is the diameter of a constellation] |
| Observation | Its diameter, remarkably, is greater than 10 AU (1.5 billion kilometers!), large enough to fill the entire inner solar system almost as far out as Jupiter. |
| Thought | A constellation is usually 10 AU in diameter. I need to convert it into light years. |
| Action | search[10 AU to light years] |
| Observation | 0.000158125 |
| Thought | A constellation is usually 0.000158125 light years in diameter, which is much smaller than 500 light years. I should search the diameter of the solar system next. |
| Action | Sedna is three times farther away from Earth than Pluto, making it the most distant observable object known in the solar system. It is 143.73 billion km from the Sun, thus giving the Solar System a diameter of 287.46 billion km. |
| Thought | The solar system is 287.46 billion km in diameter. I need to convert it into light years. |
| Action | search[287.46 billion km to light years] |
| Observation | 0.0303845459748716 |
| Thought | A constellation is usually 0.0303845459748716 light years in diameter, which is much smaller than 500 light years. Given all the information about diameters, the diameter of a galaxy is closest to 500 light years. So the answer is A. |
| Action | finish[A] |
| Observation | Episode finished, reward = 1 |

Strategy QA Prompts

---

Input-Output(IO)

| | |
|---|---|
| Question | Yes or no:  Do the anchors on Rede Globo speak Chinese? |
| Answer | no |
| | |
| Question | Yes or no:  Do the anchors on Rede Globo speak Chinese? |
| Answer | no |
| | |
| Question | Yes or no:  Would a student of the class of 2017 have amnesia about 9/11? |
| Answer | yes |
| | |
| Question | Yes or no:  Is average number of peas in a pod enough commas for a billion? |
| Answer | yes |

---

Chain-of-Thoughts (CoT)

| | |
|---|---|
| Question | Yes or no:  Do the anchors on Rede Globo speak Chinese? |
| Thought | The anchors on Rede Globo, a Brazilian television network, primarily speak Portuguese as that is the official language of Brazil.  They may have proficiency in other languages, but Chinese is not likely to be one of the languages commonly spoken by the anchors on Rede Globo.  So the answer is no. |
| Answer | no |
| | |
| Question | Yes or no:  Will the Albany in Georgia reach a hundred thousand occupants before the one in New York? |
| Thought | As of the most recent population estimates, Albany, New York, had a population of approximately 97,000 residents, while Albany, Georgia, had a population of around 73,000 residents.  Albany, New York, is the capital of the state and is a major center for business, education, and government.  It has a long history and economic significance, which attracts people to live and work in the area.On the other hand, Albany, Georgia, while an important regional center, is a smaller city in comparison.  It does not have the same level of economic or cultural influence as Albany, New York.  In conclusion, based on the current population figures and the different dynamics at play, it is unlikely that Albany, Georgia, will reach a population of one hundred thousand before Albany, New York.  So the answer is no. |
| Answer | no |
| | |
| Question | Yes or no:  Is average number of peas in a pod enough commas for a billion? |
| Thought | Generally, a typical pea pod contains around 6 to 9 peas.  A billion is a very large number 1,000,000,000 that requires 3 commas, which is less than the average number of peas in a pod.  So the answer is yes. |
| Answer | yes |
| | |
| Question | Yes or no:  Is the language used in Saint Vincent and the Grenadines rooted in English? |
| Thought | Saint Vincent and the Grenadines were once British colonies, and English became the dominant language during the colonial period.  After gaining independence in 1979, English remained as the official language of the country, and it has continued to be used in education, government, media, and daily communication.  English has permeated various aspects of society and is widely spoken by the population, though local dialects and accents may influence the spoken form of English in the region.  So the answer is yes. |
| Answer | yes |

---

ReAct

| | |
|---|---|
| | Solve a question answering task with interleaving Thought, Action, Observation steps.  Thought can reason about the current situation, and Action can be two types: |
| | (1) search[question], which searches a question on Google and returns a short snippet containing the answer.  Note that sometimes the snippet does not contain the answer, and some alternative search might be needed. |
| | (2) finish[answer], which returns the answer and finishes the task. |
| | Here are some examples. |
| | |
| Question | Yes or no:  Will the Albany in Georgia reach a hundred thousand occupants before the one in New York? |
| Thought | I need to first find the population of Albany, Georgia, then find the population of Albany, New York, then compare them. |
| Action | search[what is the current population of Albany, Georgia?] |
| Observation | The current population of Albany, Georgia is 68,181 based on our projections of the latest US Census estimates.The last official US Census in 2020 recorded ... |
| Thought | Albany, Georgia has 68,181 occupants in 2020. |
| Action | search[what is the current population of Albany, New York?] |

*Continued on next page*

Strategy QA Prompts – Continued from previous page

| | |
|---|---|
| Observation | The current population of Albany, New York is 97,593 based on our projections of the latest US Census estimates.The last official US Census in 2020 recorded ... |
| Thought | Albany, New York has 97,593 occupants in 2020, which is larger than Albany, Georgia. So Albany in Georgia will not reach a hundred thousand occupants before the one in New York, the answer is no. |
| Action | finish[no] |
| Observation | Episode finished, reward = 1 |
| | |
| Question | Yes or no: Do the anchors on Rede Globo speak Chinese? |
| Thought | I need to know what is Rede Globo first. |
| Action | search[what is Rede Globo?] |
| Observation | TV Globo formerly known as Rede Globo, is a Brazilian free-to-air television network, launched by media proprietor Roberto Marinho on 26 April 1965. |
| Thought | Rede Globo is a Brazilian television network, and Brazil is not a Chinese-speaking country, so anchors on Rede Globo do not speak Chinese. |
| Action | finish[no] |
| Observation | Episode finished, reward = 1 |
| | |
| Question | Yes or no: Would a student of the class of 2017 have amnesia about 9/11? |
| Thought | The student's awareness about 9/11 would depend on their age at the time of the event, and if the age is too young, they would not have direct memory of the event. So, I need to first know how old is a student from class of 2017 |
| Action | search[when is a student from class of 2017 born?] |
| Observation | The graduates of the class of 2017 were mostly born in 1999. Here's a look at the world they came up in. They are as old as The Phantom Menace. Midichlorians, Jar-Jar and pod racing have always been part of the Star Wars world for them. |
| Thought | If a student is born around 1999, they would have been around 2 years old during the 9/11 attacks in 2001. I need to what age would have amnesia. |
| Action | search[what age would have amnesia?] |
| Observation | Although infants use their memories to learn new information, few adults can remember events in their lives that happened prior to the age of three. Psychologists at Emory University have now documented that age seven is when these earliest memories tend to fade into oblivion, a phenomenon known as "childhood amnesia." |
| Thought | Amnesia happens for events prior to the age of three, so a student of the class of 2017 would have amnesia about 9/11. |
| Action | finish[yes] |
| Observation | Episode finished, reward = 1 |
| | |
| Question | Yes or no: Is average number of peas in a pod enough commas for a billion? |
| Thought | I need to know the average number of peas in a pod, and the number of commas for a billion, then compare them. |
| Action | search[what is the average number of peas in a pod?] |
| Observation | Every pea seed grows into a single pea plant. An average pea plant will have 6 pods with 8 peas per pod, or 48 peas in total. |
| Thought | The average number of peas in a pod is 8. I need to know how many commas in a billion. |
| Action | search[how many commas in a billion?] |
| Observation | A billion is expressed as '1,000,000,000', which amounts to three commas. |
| Thought | The average number of peas in a pod (8 peas) is indeed greater than the number of commas used in a billion (3 commas), so the answer is yes. |
| Action | finish[yes] |
| Observation | Episode finished, reward = 1. |

