# OpenReview forum: "FireAct: Toward Language Agent Finetuning"
_ICLR.cc/2024/Conference — Submitted to ICLR 2024_

### Official Review · Reviewer_3cXG · 2023-10-23

**Soundness:** 2 fair
**Presentation:** 3 good
**Contribution:** 2 fair
**Rating:** 3
**Confidence:** 5

**Summary:**

This paper introduces FireAct, a language agent that involves fine-tuning. The proposed method introduces how to leverage a strong LM with few-shot prompting to generate trajectories for fine-tuning. Experimental results demonstrate that the proposed method can improve the performance of language agents and reduce the gap between open-source LLMs and ChatGPT / GPT-4.

**Strengths:**

This paper highlights the importance of fine-tuning to obtain better language agents. Massive experimental results also explore different aspects to improve agents, including scaling effects, robustness, generalization, efficiency and cost.

**Weaknesses:**

1. Generally, I think the main contribution of this paper can be considered as a data augmentation, which aims to utilize strong LLMs to generate trajectories for language models and mix multiple training tasks for fine-tuning. The idea of mixing multiple training tasks is not interesting, and previous works like T5 or instruction tuning also adopted such an experience for tuning.
2. Although the authors have provided massive experiments to verify different fine-tuning factors in this paper, most of the conclusions are obvious and not inspired. For example like the experiments in sec 5.3, many conclusions are straightforward and it is evident that full-model training is better than LoRA. How to design experiments to prove the innovation of the proposed method (FireAct) is more important, rather than analyzing these factors.
3. Most of the experiments are only employed on three datasets (HotpotQA, StrategyQA, and MMLU). To verify the effectiveness of agents, more real-world scenarios like interactive environments are required.

Other suggestions:
1. Too many fine-tuning details and settings are given in the appendix, not the main paper. And many useless experimental results cost too many pages in the whole paper. However, these experiments do not bring any insights. On the contrary, some fine-tuning details are more important for us to know the contribution of this paper.

**Questions:**

1. The papers conduct experiments to validate the robustness to noisy tools. However, previous works (e.g., HuggingGPT, Chameleon) have adopted retrieval-based methods to utilize tools based on ChatGPT or GPT-4 and achieve some performance. So how fine-tuning-based methods compare with retrieval-based methods?
2. What is the content of Appendix A.2?

---

> ### Author Response · Authors · 2023-11-23
>
> Thank you for your detailed feedback and constructive criticism. We would like to address your concerns as follows:
>
> **On the Contribution of FiReAct:**
> While we acknowledge that the concept of data augmentation and multi-task fine-tuning is not novel in itself, the unique contribution of FiReAct lies in its application to language agent fine-tuning. We are among the first to systematically explore how agents can acquire generalizable skills through carefully designed fine-tuning trajectories. Our findings offer preliminary yet promising insights into the potential of language agent fine-tuning, paving the way for future research in this area.
>
> **Regarding Experiment Design and Conclusions:**
> We understand your concern about the apparent straightforwardness of some conclusions, such as the comparison between full-model training and LoRA. However, these results are crucial for practitioners in the field, providing concrete, quantitative insights into the performance-efficiency tradeoff in agent fine-tuning. For instance, our findings on the comparative efficiency of LoRA versus full-model training on different hardware setups (detailed in Appendix A.5) are valuable for those implementing these methods in practice.
>
> **On the Choice of Datasets:**
> We have expanded our experiments to include tasks beyond QA, as detailed in our General Response. These additional tasks encompass a broader range of real-world scenarios, further validating the effectiveness of our approach.
>
> **Concerning the Detailedness of the Appendix:**
> We appreciate your feedback on the balance between the main paper and the appendix. We invite specific suggestions on which experimental results you find less insightful and which fine-tuning details you believe are crucial but missing. Such feedback will be invaluable in refining our paper.
>
> **Comparison with Retrieval-Based Methods:**
> Your question regarding the comparison between fine-tuning-based and retrieval-based methods is insightful. It's important to note that the concept of an "agent" in our context goes beyond mere retrieval; it encompasses reasoning and tool usage in a broader sense. Our approach, therefore, offers a more comprehensive improvement of agent capabilities, as evidenced by our inclusion of tasks like Operating System, Database, and Web Browsing from AgentBench mentioned in General Response.
>
> **Content of Appendix A.2:**
> Appendix A.2 presents a detailed comparison of results across different tasks and methods, focusing on robustness analysis. We have made efforts to enhance the clarity of this section to avoid any ambiguity.

---

### Official Review · Reviewer_iJYa · 2023-10-31

**Soundness:** 3 good
**Presentation:** 3 good
**Contribution:** 2 fair
**Rating:** 6
**Confidence:** 5

**Summary:**

This paper proposes FireAct in the ReAct framework to finetune LMs with agent trajectories generated by GPT-4. FireAct enhances the training process by combining ReAct with COT and Reflexion prompting techniques, resulting in the generation of more diverse and improved training samples. Notably, FireAct does not require a few-shot prompting example during inference. The study presents several intriguing findings and highlights unexplored research questions, including the intricate relationship between the base LM and fine-tuning trajectory data, assessing the robustness of language agents, optimizing their task-solving strategies, and evaluating the effectiveness of strategy selection. In the experimental evaluation, the authors demonstrate the effectiveness and efficiency of the proposed method on various QA tasks, such as HotpotQA, Bamboogle, StrategyQA, and MMLU.

**Strengths:**

1. The motivation of this paper to systematically investigate the effect of finetuning language agents is of importance.
2. The authors have performed fine-tuning on various backbone models, including LLaMA and GPT-3.5, thereby enhancing the soundness of this study.
3. This paper uncovers valuable insights, such as the increased robustness of fine-tuned agents compared to zero-shot ones, and the ability of LLMs fine-tuned on multi-method training data to implicitly select reasoning methods.

**Weaknesses:**

1. The FiReACT method is not novel today. Several works have proposed fine-tuning Language Model Models (LLMs) on ReACT, COT, or Reflection data. Examples include "Large Language Models Are Reasoning Teachers" (Ho et al., 2022) and toolLLaMA (Qin et al., 2023).

2. The authors have not conducted enough experiments on the setting of full fine-tuning. I wonder whether the conclusions of this paper still hold when the method is applied to fully fine-tune an agent.

**Questions:**

1. What is the reason behind the observation that the fine-tuned agents exhibit more robustness compared to the zero-shot ones, and that the LLMs fine-tuned on the multi-method training data can implicitly select reasoning methods?

---

> ### Author Response · Authors · 2023-11-23
>
> We appreciate your insightful observations and questions, which allow us to further clarify the contributions of our work.
>
> **On the Novelty of FiReAct:**
> We acknowledge the existence of concurrent works such as those by Ho et al. (2022) and Qin et al. (2023), which we have duly cited and discussed in our related work section. FiReAct, however, distinguishes itself by exploring the multi-task, multi-method fine-tuning approach for language agents. Our work delves into new dimensions such as generalization to novel tasks, robustness against noise, and efficiency in agent operation, thereby contributing unique insights to the field.
>
> **Experiments on Full Fine-Tuning:**
> Our experimental setup primarily utilized LoRA fine-tuning due to computational constraints. However, we posit that our key findings, particularly regarding the enhanced robustness of fine-tuned agents to tool noise and the efficiency benefits during inference, would likely remain valid in a full fine-tuning scenario.
>
> **Robustness and Implicit Method Selection:**
> The observed robustness of fine-tuned agents, especially in comparison to zero-shot counterparts, can be attributed to the inclusion of noisy data in the fine-tuning trajectories. This exposure enables the agents to better handle imperfect tool outputs, a scenario less effectively addressed by few-shot prompting, which often relies on idealized examples (refer to Table 3 for details).
>
> The phenomenon of implicit method selection in multi-method fine-tuned agents is indeed intriguing. Our current hypothesis is that the fine-tuning process, which selectively focuses on successful trajectories, results in a varied distribution of training questions associated with different methods. This could enable the fine-tuned agents to learn associations between question features and the most effective method for addressing them. We believe this presents an exciting avenue for future research to explore in depth.

---

### Official Review · Reviewer_6Re8 · 2023-11-01

**Soundness:** 2 fair
**Presentation:** 2 fair
**Contribution:** 2 fair
**Rating:** 3
**Confidence:** 3

**Summary:**

This study delves into the relatively under-researched area of refining language models (LMs) to create linguistic agents. By employing a straightforward, regulated framework that utilizes a Google search API for question answering (QA), this paper conducts a thorough examination across a range of foundational LMs, agent strategies, data used for fine-tuning, and QA challenges. This investigative work uncovers new understanding regarding how the size of the base LM and the fine-tuning dataset influences outcomes, the integration of path data from diverse tasks and agent techniques, and the resilience of these systems to various forms of data disturbances.

**Strengths:**

Pros:
1. This paper claims that they study a new direction: Agent fine-tuning.
2. Comprehensive experiments show the effectiveness of the proposed Agent fine-tuning.
3. Detailed analyses are given to illustrate Agent fine-tuning.

**Weaknesses:**

Cons:
1. It is unclear the advantage of FiReAc compared to ReAct considering FiReAc is fine-tuned based on ReAct trajectories. The authors claimed that “FiReAct agents benefit from the diversity of learning support, thus become more robust to external noises, more generalizable to novel tasks, and more flexible to choose various agent methods and task solving strategies adaptive to the problem at hand.” However, intuitively, it is hard to understand the reasons why fine-tuning can make the agents more robust and generalizable. It is suggested that authors could provide more explanations.
2. It is known that fine-tuning can hurt the generalization ability of LLMs. But the authors claimed that the proposed “agent fine-tuning” method is “more generalizable to novel tasks”, which is hard to understand.
3. The method seems to be very trivial. It seems this paper just renames the conventional “fine-tuning” as “agent fine-tuning”.
4. The technical contribution is limited because It is unclear what is technical challenges of “agent fine-tuning”. It Is suggested the authors provide more explanations.
5. It is suggested the authors summarize their contributions more clearly. It is very vague to say “To sum up our contributions, we advocate for the overlooked direction of language agent fine-tuning, and propose novel methodologies, systematic experiment designs, practical insights and guidelines, as well as new research questions for this direction."

**Questions:**

See weaknesses.

---

> ### Author Response · Authors · 2023-11-23
>
> We are grateful for your detailed feedback and the opportunity to clarify aspects of our work.
>
> **Advantages of FiReAct over ReAct:**
> Your query about the benefits of FiReAct compared to ReAct is crucial. FiReAct's fine-tuning process, unlike ReAct's few-shot prompting, involves learning from a broader range of trajectories that encompass diverse task-solving strategies. This diversity is key to FiReAct's enhanced robustness and generalizability. Specifically:
> - FiReAct shows improved performance on the test sets of training tasks (refer to Table 2).
> - When applied to novel tasks, FiReAct maintains or enhances performance, as evidenced in Table 3 and our General Response. This is a notable departure from traditional NLP fine-tuning, where the focus is often on specific facts or task formats. FiReAct, by teaching LLMs how to reason and utilize tools like Google search, imparts skills that are transferable across various tasks.
> - FiReAct demonstrates greater resilience to noise in tool outputs. This is because the fine-tuning process includes trajectories with noise, teaching the model to handle imperfect data. In contrast, few-shot prompting often relies on 'ideal' examples, which may not prepare the model for real-world variability (see Table 3).
>
> **On the Alleged Triviality of Agent Fine-Tuning:**
> We appreciate your concern regarding the perceived simplicity of our approach. However, the novelty of FiReAct lies in:
> - Addressing the gap in research: Prior work has largely overlooked fine-tuning LLMs specifically for agent roles, with most studies either fine-tuning LLMs for non-agent tasks or using prompting for agent tasks.
> - Demonstrating multi-faceted empirical benefits: FiReAct, despite its simplicity, showcases significant benefits in terms of robustness, generalization, efficiency, and cost, which are crucial for practitioners in the field of agent development.
> - Exploring new technical depths: FiReAct pioneers the approach of multi-method, multi-task fine-tuning for agents. It demonstrates that even smaller LLMs can effectively acquire generalizable agent skills through supervised fine-tuning, presenting a viable alternative to more complex and less controllable methods like reinforcement learning.
>
> **Clarification of Contributions:**
> In response to your suggestion, we have revised our introduction to more clearly articulate our contributions. We now explicitly state how FiReAct advances the field of language agent fine-tuning, detailing the novel methodologies, systematic experiment designs, practical insights, and new research questions that our work introduces.

---

### Official Review · Reviewer_GhVh · 2023-11-02

**Soundness:** 3 good
**Presentation:** 3 good
**Contribution:** 2 fair
**Rating:** 5
**Confidence:** 4

**Summary:**

The paper introduces "FiReAct," a method for fine-tuning language models to function as language agents capable of reasoning and interacting with external environments. It challenges the conventional few-shot prompting approach, suggesting that fine-tuning with agent trajectories is more robust and efficient. The study uses a controlled environment with a Google search API for question answering to demonstrate the benefits of FiReAct. Key contributions include a novel fine-tuning methodology, a set of empirical guidelines for implementing such agents, and the release of code and model checkpoints for future research. The authors argue for fine-tuning over prompting when data and task understanding permit, setting new directions for language agent development.

**Strengths:**

The strengths of the paper, as identified through the experimental results, include the following:

1. FiReAct agents eliminate the need for few-shot prompting examples during inference, which results in cost savings, faster operation, and increased convenience compared to prompting-based agents.
2. These agents benefit from a diverse learning support, which enhances their robustness against external noise and improves their generalizability to novel tasks.
3. FiReAct provides flexibility for agents to choose from various methods and strategies for solving tasks, allowing them to adapt to the problem at hand more effectively.

Moreover, the paper opens up new avenues for research by highlighting previously unexplored questions related to the interactions between the base language model (LM) and the fine-tuning data, evaluating the robustness and strategic decision-making of language agents, and the systematic analysis of fine-tuning data design for language agents.

**Weaknesses:**

Task and Tool Limitation: The research is limited to question-answering (QA) tasks and relies on a single tool (Google search) for evaluation. This limitation questions the generalizability of FiReAct to other types of tasks and tools, which is a critical aspect for language agents intended for broader applications. To make the conclusions more convincing, it would be great if you could evaluate on other types of interactive tasks such as ScienceWorld, Mind2Web, InterCode, etc.

Methodology Limitations: The paper focuses on three methods that maintain a single autoregressive trajectory context, which might not be representative of more complex agent behaviors. The ability to handle multiple prompts, roles, and contexts is a significant aspect of agent design that is not addressed in the current research.

Multi-Task Constraints: The multi-task setup is confined to only three QA tasks, and the most advanced language model (LM) fine-tuned is GPT-3.5. The limited scope in both the variety of tasks and the LM capabilities may restrict the insights into the potential of FiReAct for scaling up to more complex multi-task environments.

**Questions:**

1. How do the authors envision the application of FiReAct to tasks beyond QA and the use of tools other than Google search? Can FiReAct work on other types of interactive tasks such as ScienceWorld, Mind2Web, InterCode, etc? Why do you choose to focus on QA only?
2. Can the authors discuss potential approaches for fine-tuning more advanced agents involving multiple prompts, roles, and contexts?
3. How might prompting and fine-tuning be optimally combined in a complex agent system?
4. What are the anticipated difficulties in scaling FiReAct to a large-scale multi-task environment?
5. How might newer or more powerful models like GPT-4 affect the outcomes, and what would be the implications for the FiReAct methodology?

---

> ### Author Response · Authors · 2023-11-23
>
> We sincerely appreciate your insightful feedback and constructive suggestions. Here is our response to the key points raised:
>
> **Task and Tool Limitation:**
> We acknowledge your concern regarding the generalizability of FiReAct. In response, we have expanded our experiments to include additional tasks from AgentBench, which extends beyond QA and incorporates a variety of tools. These new experiments are designed to demonstrate FiReAct's adaptability to a broader range of tasks and tools.
>
> **Methodology Limitations:**
> We understand the importance of handling multiple prompts, roles, and contexts in agent design. Our current focus on methods like CoT, ReAct, and Reflexion, which align with the ReAct autoregressive format, was a strategic choice due to their compatibility with multi-method fine-tuning. However, we recognize this as a limitation and an exciting avenue for future research, as noted in our revised manuscript.
>
> **Multi-Task Constraints:**
> For further details on our expansion beyond QA tasks, please refer to our General Response. Regarding the use of GPT-3.5, it represents the most advanced publicly available LM for fine-tuning at the time of our study. We are actively exploring the integration of more diverse tasks and advanced LMs in our ongoing research.
>
> **Application of FiReAct Beyond QA and Advanced Agents:**
> As detailed in our General Response, FiReAct's application extends to various interactive tasks, including those mentioned. Our initial focus on QA was due to its well-defined structure and data availability, making it an ideal starting point for demonstrating FiReAct's capabilities.
>
> **Combining Prompting and Fine-Tuning:**
> As discussed in Section 8 of our paper, we advocate for a balanced approach. Fine-tuning is preferable for well-defined, frequently used tasks with sufficient data, while prompting offers flexibility and convenience for less defined or zero-shot scenarios.
>
> **Scaling FiReAct to Multi-Task Environments:**
> Our initial experiments do not indicate a performance decrease when scaling FiReAct to more tasks. Future work will investigate the upper limits of task complexity and quantity that FiReAct can effectively manage.
>
> **Impact of Newer Models like GPT-4:**
> Our experiments across various models, including Llama-2 and CodeLlama, demonstrate FiReAct's effectiveness irrespective of model size or family. We anticipate that fine-tuning with GPT-4 would enhance performance, generalizability, and robustness, and we look forward to exploring this in future work.

---

> > ### Comment · Reviewer_GhVh · 2023-12-04
> >
> > Thanks for the responses. I have read the other reviews and the authors' rebuttal. Considering the overall quality of the paper, I'd like to keep my score.

---

### Author Response · Authors · 2023-11-23
**General Response to Reviewers**

We would like to extend our heartfelt thanks to all the reviewers for their insightful comments on our paper. Your feedback is invaluable in enhancing the quality of our work. Below, we address the related comments collectively in a general response:

**Point #1: Experiments on Additional Agent Tasks**

In response to concerns about the limited scope of task and tool coverage, we have expanded our framework to include three additional tasks from AgentBench (Liu et al., 2023): Operating System, Database, and Web Browsing. We adapted the GPT-3.5-Turbo-1106 model using the FireAct approach and conducted ablation studies in two settings: 1) [Task]-only: FireAct model trained exclusively on the specified task; 2) w/o [Task]: FireAct model trained on all tasks except the specified one.

|               | OS    | DB    | WB  |
|---------------|-------|-------|-----|
| GPT-3.5-Turbo | 32.6  | 36.7  | 20  |
| Multi-Task FireAct | 41 | 47.6 | 28  |
| **Ablations** |       |       |     |
| OS-only       | 30.5  |       |     |
| w/o OS        | 30.5  |       |     |
| DB-only       |       | 50    |     |
| w/o DB        |       | 41    |     |
| WB-only       |       |       | 16  |
| w/o WB        |       |       | 23  |

These results demonstrate that the FireAct framework, when augmented for multi-tasking, is versatile and applicable to a range of agent tasks beyond complex QA scenarios like HotpotQA. The ablation studies further suggest that multi-task training with FireAct generally yields superior results, highlighting a promising avenue for combining tasks and implementing regularization in language agent fine-tuning.

**Point #2: Summary of Revision**

Please Note: In our revised rebuttal document, we have highlighted all modifications and additions in blue for easy identification and reference.

*Methodology Limitations:* Our paper initially focused on methods maintaining a single autoregressive trajectory context, which may not fully represent complex agent behaviors. Acknowledging this, we have now highlighted it as a limitation and an exciting area for future exploration in language agent fine-tuning.

*Performance and Task Complexity:* We observed no performance degradation when applying FireAct across multiple tasks. Future work will explore the upper limits of task quantity and complexity manageable by FireAct, incorporating insights from the GR agentbench results.

*Clarification of Contributions:* We have revised our introduction to more explicitly summarize our contributions, moving away from vague statements to specifically highlight our advocacy for language agent fine-tuning, our novel methodologies, systematic experiment designs, practical insights, guidelines, and the new research questions we have introduced in this field.

*Appendix and Robustness Analysis:* In Appendix A.2, Table 9 now offers a more detailed comparison of results across different tasks and methods, particularly focusing on robustness analysis. We have enhanced the clarity of our Appendix to minimize ambiguity.

---

### Meta-Review · Area_Chair_eG4t · 2023-12-03

**Metareview:**

The paper proposes fine-tuning LLMs into language agents with reasoning and interaction capabilities. It argues that fine-tuning with agent trajectories is superior to the traditional few-shot prompting technique, offering improved robustness and efficiency. The study utilizes a controlled environment based on Google search to analyze various aspects of agent fine-tuning, including the impact of base LM size, data sources, and different tasks. It leverages GPT-4-generated trajectories for training and combines ReAct with CoT and Reflexion prompting to diversify training samples. Although the method shows promising results, multiple reviewers raised concerns about the significance of the contributions, given the unsurprisingly expected results, and the generalizability of the experiments. The concerns have not been fully addressed during rebuttal. Thus, I recommend rejection of the paper. The authors are encouraged to improve the paper based on the reviewers' comments and resubmit to a future venue.

**Justification For Why Not Higher Score:**

Unresolved concerns regarding the contribution significance and generalizability of the experiments

**Justification For Why Not Lower Score:**

N/A

---

### Decision · Program_Chairs · 2024-01-16

Reject